# Multiscale representations of community structures in attractor neural networks

**Tatsuya Haga** *, **Tomoki Fukai** *

Okinawa Institute of Science and Technology, Onna-son, Okinawa, Japan

* tatsuya.haga3@oist.jp(TH); tomoki.fukai@oist.jp(TF)

**Data Availability Statement:** All relevant data are within the paper and its Supporting Information files. All codes for simulations and analyses are available at Zenodo (http://doi.org/10.5281/zenodo. 5210905). Images used for the image

## Abstract

Our cognition relies on the ability of the brain to segment hierarchically structured events on multiple scales. Recent evidence suggests that the brain performs this event segmentation based on the structure of state-transition graphs behind sequential experiences. However, the underlying circuit mechanisms are poorly understood. In this paper we propose an extended attractor network model for graph-based hierarchical computation which we call the Laplacian associative memory. This model generates multiscale representations for communities (clusters) of associative links between memory items, and the scale is regulated by the heterogenous modulation of inhibitory circuits. We analytically and numerically show that these representations correspond to graph Laplacian eigenvectors, a popular method for graph segmentation and dimensionality reduction. Finally, we demonstrate that our model exhibits chunked sequential activity patterns resembling hippocampal theta sequences. Our model connects graph theory and attractor dynamics to provide a biologically plausible mechanism for abstraction in the brain.

## Author summary

Our experiences are often hierarchically organized, so is our knowledge. Identifying meaningful segments in hierarchically structured information is crucial for many cognitive functions including visual, auditory, motor, memory, language processing, and reasoning. Herein, we show that the attractor dynamics of recurrent neural circuits offer a biologically plausible way for hierarchical segmentation. We found that an extended model of associative memory autonomously performs segmentation by finding groups of tightly linked memories. We proved that the neural dynamics of our model mathematically coincide with optimal graph segmentation in graph theory and are consistent with the experimentally observed nature of human behaviors and neural activities. Our model established a previously unexpected relationship between attractor neural networks and the graph-theoretic processing of knowledge structures. Our model also provides experimentally testable predictions, particularly regarding the role of inhibitory circuits in controlling representational granularity.

segmentation task were not included but URLs of images are given with published codes.

**Funding:** This work was partially supported by Kakenhi nos. 18H05213 and 19H04994 to TF from Japan Society for the Promotion of Science, Japan (https:// www.jsps.go.jp/english/index.html), and Kakenhi no. 21K15611 to TH from Japan Society for the Promotion of Science, Japan (https://www. jsps.go.jp/english/index.html). The funders had no role in study design, data collection and analysis, decision to publish, or preparation of the manuscript.

**Competing interests:** The authors have declared that no competing interests exist.

# Introduction

The brain builds a hierarchical knowledge structure through the abstraction of conceptual building blocks such as groups and segments. This ability of the brain is essential for various cognitive functions such as chunking of items, which increases the number of items retained in a limited capacity of working memory [1], segmentation of words, which is essential for learning and comprehension of language [2–4], and temporal abstraction of repeated sequential actions, which accelerates reinforcement learning [5].

Experimental evidence suggests that the brain performs segmentation based on the graph structures behind the experiences. When the brain experiences a sequence of events, it learns the temporal associations between the successive events and eventually captures the structure of the state-transition graph behind the experience. It has been shown that event segmentation performed by human subjects behaviorally reflects community structures (or clusters) of such state-transition graphs, and neurobiologically, sensory events within the same community are represented by more similar activity patterns than those belonging to other communities [6,7]. Such graph segmentation of events is considered to benefit the temporal abstraction of actions in reinforcement learning [8,9]. Furthermore, graph-based representations can explain many characteristics of place cells and entorhinal grid cells [10].

Despite its behavioral and representational evidence, the biological mechanism that creates graph-based representations remains unknown. Conventionally, circuit-level mechanisms in hippocampal and cortical processing have been modeled as attractor-based associative memory networks [11–13]. Experiments have revealed some hallmarks of associative memory networks such as Hebbian learning (as spike-timing-dependent plasticity) [14,15], pattern completion and attractor states [16–19] in the brain. In the context of associative memory, temporal associations between successive events can be modeled as hetero-associations between successively activated cell assemblies through Hebbian learning [20–23]. This learning scheme creates correlated attractors from uncorrelated stimuli. Correlations depend on the temporal distance between the memorized events along the event sequence, which quantitatively agrees with neural recordings from the monkey brain [24,25]. Such correlated attractors, and hence the class of associative memory models, can be potentially extended to offer a biologically plausible representational basis for more general graphical structures. However, this hypothesis has not been examined.

In this study, we propose a generalized class of associative memory networks [20–23] that performs graph-based segmentation and abstraction. Herein, we present two major extensions. First, we generalize the one-dimensional sequential structure of temporal associations in the conventional model [20–23] to arbitrary symmetric graphs. Second, we allow the model to have negative associative weights which can be interpreted as assembly-specific inhibition [26]. We found that this network generates mixed representations that are shared by multiple memory items within the same communities in the graph, which fits with human experiments [6,7]. We mathematically revealed that information representations in the attractor state in our model are related to graph Laplacian eigenvectors, a popular mathematical method for graph segmentation [27,28] and nonlinear dimensionality reduction [29]. Because of this property, we call our model Laplacian associative memory (LAM), and demonstrate that LAM applies to problems related to graph Laplacian eigenvectors such as subgoal finding in reinforcement learning [8,9]. Our model predicts that the scale of the representations (the size of the represented communities) is modulated by the relative strength of local and global inhibitory circuits, which indicates an active role of target-specific inhibition [26] and inhomogeneous neuromodulation of inhibitory circuits [30,31]. Finally, we show that LAM with asymmetric links generates chunked sequential activities observed in the hippocampus

[32,33]. Our model establishes a theoretical relationship between associative memory networks and graph theory, providing a biologically plausible dynamical mechanism for hierarchical abstraction in the brain.

## Results

### The Laplacian associative memory model

Laplacian associative memory (LAM) is a novel class of Hopfield-type recurrent network models [11–13,20,23]. Let us define a network of $N$ units $x_i(t)$ ($i = 1, \cdots, N$; $0 \leq x_i(t) \leq 1$) as follows:

$$\dot{x}_i = -x_i + \Theta\left(\sum_{j=1}^{N} w_{ij} x_j\right), \tag{1}$$

where $w_{ij}$ is the synaptic weight and $\Theta(x)$ is a step function ($\Theta(x) = 1$ if $x > 0$, otherwise $\Theta(x) = 0$). We assume that each memory item (e.g. sensory stimuli, places or events) is represented by a 0–1 binary random memory pattern $\xi_i^\mu$ ($i = 1, \cdots, N$; $\mu = 1, \cdots, P$) with sparsity $p$ ($\mathrm{P}[\xi_i^\mu = 1] = p$). In this study, we used $N = 10000$ and $p = 0.1$ unless otherwise specified. We set the synaptic weights from these memory patterns as

$$w_{ij} = \frac{1}{NV} \sum_{\mu=1}^{P} \sum_{\nu=1}^{P} (\alpha \delta_{\mu\nu} + H_{\mu\nu}) \tilde{\xi}_i^\mu \tilde{\xi}_j^\nu - \frac{1}{N}(\alpha+1)\gamma, \tag{2}$$

where $\tilde{\xi}_i^\mu = \xi_i^\mu - P^{-1}\sum_\mu \xi_i^\mu$ and $V = p(1-p)$. The term $\alpha\delta_{\mu\nu}$ represents auto-association within each item, where $\delta_{\mu\nu}$ is the Kronecker delta and $\alpha$ is a modifiable parameter that determines the strength of auto-association. On the other hand, $H_{\mu\nu}$ is a hetero-associative weight between memory items $\mu$ and $\nu$ ($H_{\mu\mu} = 0$). Parameter $\gamma \geq 0$ provides an additional global inhibitory effect [13]. In short, this network stores multiple cell assemblies ($P$ memory patterns) through auto-associative Hebbian learning and links them through hetero-associative Hebbian learning (Fig 1, left). We construct hetero-associative weights from a normalized adjacency matrix of a state-transition graph, or generally, other graphs such as semantic relationships. Specifically, we hypothesize that the hetero-associative weight matrix $\mathbf{H} = (H_{\mu\nu})_{1\leq\mu\leq P, 1\leq\nu\leq P}$ is constructed as $\mathbf{D}^{-\frac{1}{2}}\mathbf{A}\mathbf{D}^{-\frac{1}{2}}$ (symmetric normalization) or $\mathbf{D}^{-1}\mathbf{A}$ (asymmetric normalization) where $\mathbf{A}$ and $\mathbf{D}$ are the adjacency matrix and the degree matrix of the graph, respectively. As in the graph

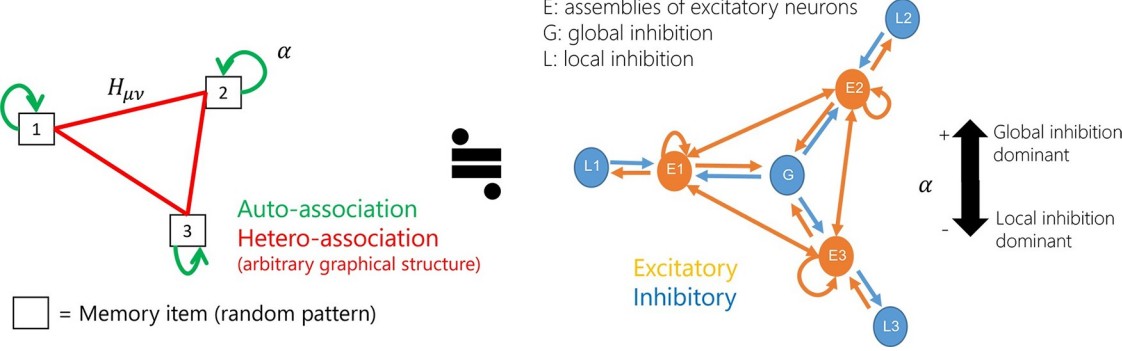

**Fig 1. Laplacian associative memory (LAM) model.** Left: associative memory network model with auto-association and hetero-association. The parameter $\alpha$ indicates the auto-associative strength. Right: Equivalent biological neural network model which contains local (assembly-specific) and global (non-specific) inhibition. The parameter $\alpha$ indicates the ratio between local and global inhibition.

Laplacian, two normalization yields the same results qualitatively. However, the symmetric normalization model enables formal theoretical analyses and the asymmetric normalization model provides a biologically plausible interpretation of the model. Asymmetrically normalized weights directly correspond to the transition probability matrix for random walk on the graph [28] and therefore can be learned in sequential experiences through Hebbian learning of successively activated cell assemblies [22,25]. Therefore, the structure of hetero-associative links is a graph reflecting the statistical structure behind experiences in which we may find some communities.

LAM can be regarded as a generalization of previous associative memory models. When $\alpha > 0$ and all $H_{\mu\nu}$ are zero, the LAM is analogous to the conventional Hopfield-type model storing biased memory patterns [11–13]. If only adjacent items are associated ($H_{\mu,\mu+1} = H_{\mu+1,\mu} > 0$, and all other $H_{\mu\nu}$ are zero) so that associative links form a one-dimensional chain, the model coincides with an associative memory model for a repeated sequence of sensory inputs [20,23]. However, unlike the previous models, LAM can also take other arbitrary hetero-associative link structures, possibly formed through sensory experiences with complex state transition structures rather than a sequential experience repeated in the same order. Furthermore, we did not restrict the parameter $\alpha$ to being positive, allowing inhibitory auto-association. We found unique behaviors of LAM mostly in the regime of negative auto-association, which has not been extensively investigated previously.

We clarify the biological interpretation of the model by the decomposition of excitatory and inhibitory components. Here, we assume asymmetric normalization of the hetero-associative weights. As in a previous study [23], we can decompose the weights as

$$w_{ij} = w_{ij}^{\mathrm{E}} - (\alpha_{\max} - \alpha)w_{ij}^{\mathrm{L}} - (1 + \alpha)w_{ij}^{\mathrm{G}}. \tag{3}$$

$$w_{ij}^{\mathrm{E}} = \frac{1}{NV} \sum_{\mu=1}^{P} \sum_{\nu=1}^{P} (\alpha_{\max}\delta_{\mu\nu} + H_{\mu\nu})\xi_i^{\mu}\xi_j^{\nu}, \tag{4}$$

$$w_{ij}^{\mathrm{L}} = \frac{1}{NV} \sum_{\mu=1}^{P} \xi_i^{\mu}\xi_j^{\mu} \tag{5}$$

$$w_{ij}^{\mathrm{G}} = \frac{P}{NV} \left(\frac{1}{P}\sum_{\mu=1}^{P}\xi_i^{\mu}\right)\left(\frac{1}{P}\sum_{\nu=1}^{P}\xi_j^{\nu}\right) + \frac{1}{N}\gamma. \tag{6}$$

Here, we used the constraint $\sum_{\nu=1}^{P} H_{\mu\nu} = 1$ and approximated $\sum_{\mu=1}^{P} H_{\mu\nu} \approx 1$. The decomposed weights $w_{ij}^{\mathrm{E}}$, $w_{ij}^{\mathrm{L}}$ and $w_{ij}^{\mathrm{G}}$ are always non-negative; thus, $w_{ij}^{\mathrm{E}}$ is an excitatory component and $-(\alpha_{\max} - \alpha)w_{ij}^{\mathrm{L}}$ and $-(1 + \alpha)w_{ij}^{\mathrm{G}}$ are inhibitory components in the range $-1 < \alpha < \alpha_{\max}$. The $w_{ij}^{\mathrm{E}}$ component can be regarded as an excitatory connection that reflects the structure of cell assemblies. The $w_{ij}^{\mathrm{L}}$ component is assembly specific local inhibition, whereas $w_{ij}^{\mathrm{G}}$ is a nonselective global connection. Therefore, LAM can be regarded as a circuit with local and global inhibition, in which the parameter $\alpha$ determines the ratio between the strengths of the two types of inhibitory circuits (Fig 1, right). Biologically, the difference in $\alpha$ may correspond to the anatomical inhomogeneity of the interneurons. Otherwise, the balance of inhibition can be changed through the inhomogeneous modulation of interneurons by acetylcholine [30,31].

## Multiscale representation of community structures in LAM

To demonstrate the representations in LAM, we tested three representative graph structures. The first is the graph used previously to study how humans segment temporal sequences obeying probabilistic state-transition rules [6,7] (Fig 2A). The second is the karate club network [34], a popular dataset for testing community detection methods in graph theory (Fig 2F). The

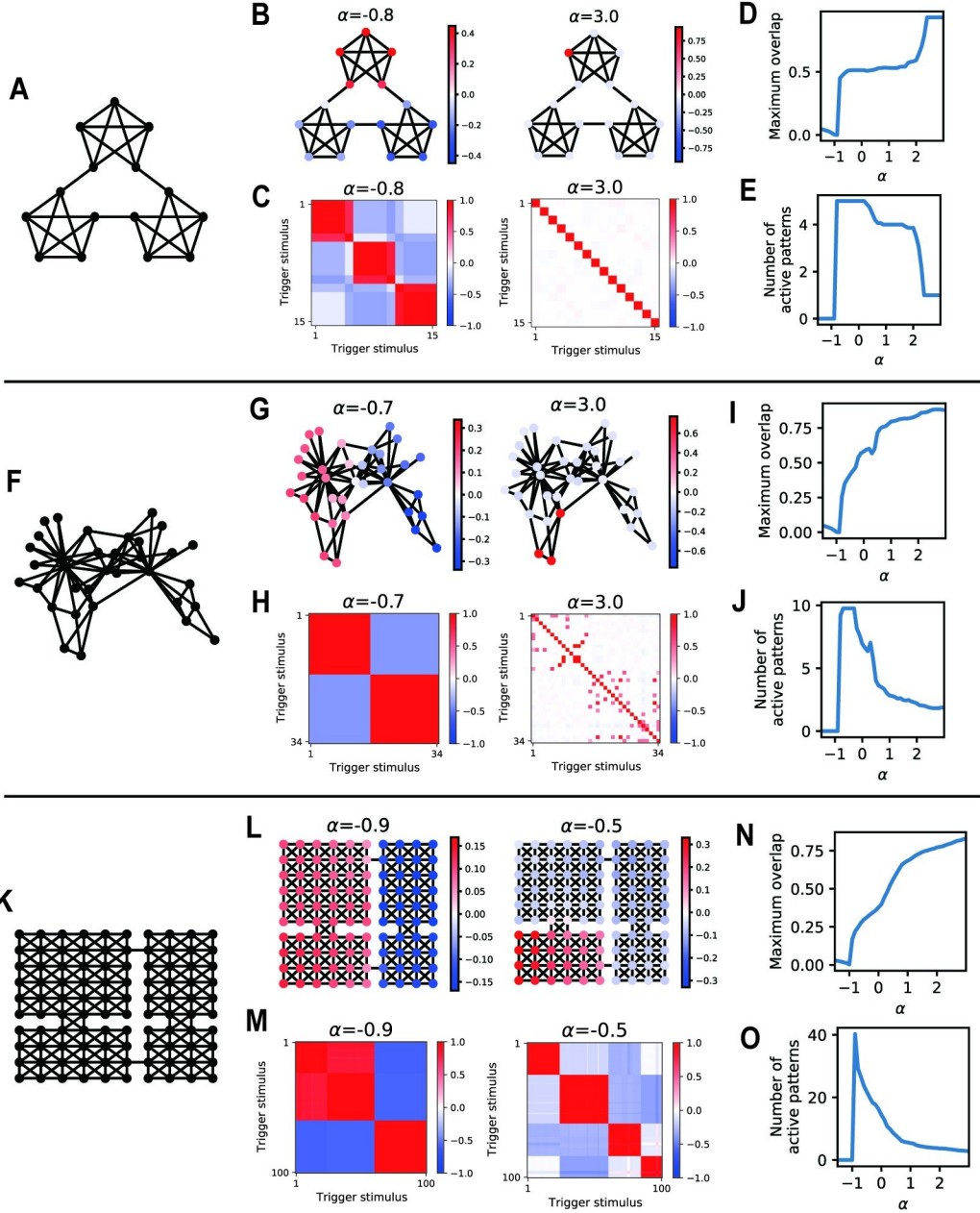

**Fig 2. LAM generates multi-scale representations for community structures.** (A) Graph used by Schapiro et al. (2013) [6]. (B) Pattern overlaps of example attractors. (C) Correlation matrices between activity patterns in the attractor states reached from different trigger stimuli (trigger nodes). (D) Maximum pattern overlaps obtained by various values of $\alpha$. (E) Numbers of active patterns obtained by various values of $\alpha$. In D and E, we averaged values from all attractors reached from different trigger stimuli. (F) Karate club network[34]. (G-J) Results for Karate-club network. (K) A compartmentalized room structure[5,9,10,36] (four-room graph). (L-O) Results for a four-room graph.

third graph represents the structure of compartmentalized rooms (Fig 2K), which is often used as the state-transition graph for reinforcement learning [5,8–10]. For each graph, we assigned a random binary pattern to each node and constructed an LAM network with hetero-associative weights $H_{\mu\nu}$ based on the adjacency matrices of the graph. This setting implicitly assumes that sensory states represented by nodes are independent from those represented by other nodes, which may represent mutually uncorrelated sensory stimuli [6,7] or assemblies of local and sparse place cells. We initialized the activity of the LAM network with one of the assigned memory patterns (trigger stimulus), and simulated the dynamics of the network for sufficiently long time until the network converged to an attractor state (S1 Fig). We regard the activity pattern at the end of each simulation as a neural representation of the node corresponding to the trigger stimulus (trigger node). For each attractor state, we calculated an index called as pattern overlap to evaluate the degree of retrieval of each memory pattern:

$$m^{\mu} = \frac{1}{NV}\sum_{i=1}^{N}\tilde{\xi}_i^{\mu}x_i. \tag{7}$$

This index measures the degree of overlap between embedded memory patterns and activity patterns in the model (large positive values indicate significant activation of the memory pattern). Pattern overlaps have been traditionally used in the analysis of memory recall in Hopfield-type models [13,20,35] because the dynamics and energy of the model can be described by the function of pattern overlaps instead of neural activities. Furthermore, we calculated the correlations between attractor patterns obtained from different trigger stimuli.

LAM converged to various attractor patterns depending on the trigger nodes and the value of auto-associative weight $\alpha$. Generally, the maximum pattern overlaps of attractors had large positive values in the parameter region $\alpha > -1$ (Fig 2D, 2I and 2N), indicating that memory recall occurred in this region. When $\alpha$ had large positive values (global inhibition was dominant), attractors locally represented one or a few nodes in the graph (Fig 2B and 2G, right) and attractor patterns reached from different trigger nodes were uncorrelated with each other (Fig 2C and 2H, right). These attractors correspond to the retrieval of individual memories observed by conventional Hopfield-type models. In contrast, when $\alpha$ is closer to -1 (local inhibition is dominant), multiple memory patterns are active in the attractor states. Quantitatively, the average number of active patterns is maximum at $\alpha \approx -1$ and decreases as $\alpha$ increases (Fig 2E, 2J and 2O; for the definition of active patterns, see Methods: Simulations of the network model). Especially at $\alpha \approx -1$, distributions of pattern overlap represent large communities in graphs (Fig 2B and 2G, left), and accordingly, the pattern correlation between attractor patterns is high within each community (Fig 2C, 2H and 2M, left). When $\alpha$ took an intermediate value, LAM represented a mesoscale community in the four-room graph (Fig 2L, right). This result demonstrates that LAM generates mixed representations for communities in hetero-associative links by partially recalling multiple memory patterns simultaneously in attractor states. Accordingly, representations (attractor patterns) for nodes within a community are highly correlated, which agrees with the results of previous experiments [6,7].

Additionally, we checked the effect of parameter settings on the model behavior in the simulation of the four-room graph. First, we changed the number of neurons $N$ from the original setting of $N = 10000$, and the model behaviors were qualitatively the same for $N = 5000$, 15000, 20000 but significantly impaired at $N = 1000$ (S2 Fig). This indicates that $N$ must be large enough for the network to work stably, which is the same property as conventional Hopfield-type models [11,12,35]. In addition, we tested the change in sparsity $p$ (0.05 and 0.2) and it did not change the qualitative results (S3 Fig).

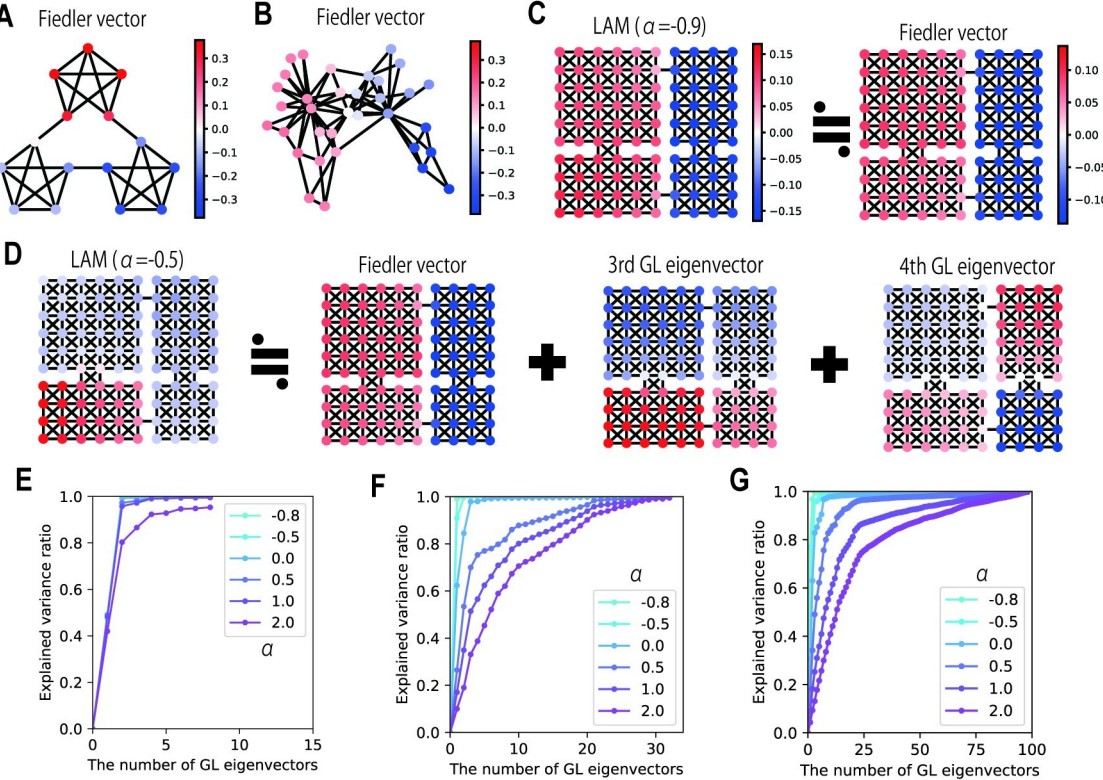

**Fig 3. The relationship between Graph Laplacian eigenvectors and LAM.** (A) Fiedler vector (GL eigenvector with the second smallest eigenvalue) for the graph in Schapiro et al. (2013). (B) Fiedler vector for karate-club network. (C) The comparison of pattern overlaps in LAM ($\alpha = -0.9$) and Fiedler vector for the four-room graph. (D) A schematic diagram showing that pattern overlaps in LAM ($\alpha = -0.5$) is mostly explained by the combination of multiple GL eigenvectors with small eigenvalues. (E-G) The explained variance ratio in linear regressions of pattern overlaps by various numbers of GL eigenvectors. The color indicates the value of $\alpha$. In each condition, we plotted the average value of the explained variance ratio of attractors reached from all trigger stimuli. (E) Results from the graph by Schapiro et al. (2013). (F) Results for the karate-club network. (G) Results for the four-room graph.

## The theoretical relationship between LAM and graph Laplacian

We analyzed the mathematical mechanism behind the representations of LAM and found that the representations are related to the graph Laplacian (GL). GL is a matrix defined from a graphical structure and its eigenvectors are used for various applications. One popular application is graph segmentation (also called community detection) because it has been shown that the signs of elements in GL eigenvectors indicate optimal two-fold segmentation of a graph [27,28] (examples are shown in Fig 3A, 3B and 3C). GL eigenvectors provide segmentation at various levels depending on their eigenvalues (a small eigenvalue corresponds to coarse resolution with large communities); thus, combinations of multiple eigenvectors provide multi-level segmentation. This property is used for image segmentation [27,28]. In another aspect, GL eigenvectors is also used for nonlinear dimensionality reduction [29], which provides low-dimensional representations for nodes (data points) in which the structure is represented through similarity. As for the connection to neural representations, GL eigenvectors become grid-like code in the homogenous space, and their distortion caused by inhomogeneity fits with the experimental observation of grid cells, and predictive spatial representations in the hippocampus can be eigendecomposed into GL eigenvectors [10]. For the definition of GL

and a brief review of its mathematical properties, see Methods: Definition and mathematical properties of graph Laplacian.

We performed a formal theoretical analysis to show the relationship between LAM and GL. Here, we use symmetric normalization of hetero-associative weights ($\mathbf{H} = \mathbf{D}^{-\frac{1}{2}}\mathbf{A}\mathbf{D}^{-\frac{1}{2}}$), which yields the same results as those shown above (S4 Fig). Then, we can define the energy function of the model as:

$$E = -\frac{1}{NV}\sum_{i=1}^{N}\sum_{j=1}^{N}w_{ij}x_i x_j. \tag{8}$$

As in the conventional Hopfield model [11], the dynamics of LAM monotonically decrease this energy (S5 Fig). Although the decrease in this energy in each step is rigorously guaranteed only when we use sequential updates of neural activities [11], updates in Eq (1) also decreases the energy at the population level because two updates fall into the same dynamical equation by mean-field approximation [23]. Considering a vector of pattern overlaps $\mathbf{m} = (m^1,\ldots,m^P)^{\mathrm{T}}$ (pattern overlap vector), and the vector rescaled by the degree matrix $\tilde{\mathbf{m}} = \mathbf{D}^{-\frac{1}{2}}\mathbf{m}$, then the energy function can be rewritten as

$$E = \tilde{\mathbf{m}}^{\mathrm{T}}\mathbf{L}\tilde{\mathbf{m}} + (\alpha + 1)[-\tilde{\mathbf{m}}^{\mathrm{T}}\mathbf{D}\tilde{\mathbf{m}} + \gamma V^{-1}(m^0)^2], \tag{9}$$

where $\mathbf{L}$ is the GL for the hetero-associative link structure (the state-transition graph), and $m^0$ is the mean activity level in the network. Here we find the minimization of $\tilde{\mathbf{m}}^{\mathrm{T}}\mathbf{L}\tilde{\mathbf{m}}$ under the constraint of $\tilde{\mathbf{m}}^{\mathrm{T}}\mathbf{D}\tilde{\mathbf{m}}$. This is the same objective as graph segmentation [27] and graph-based dimensionality reduction [29], for which GL eigenvectors provide optimal solutions. Therefore, we can expect that GL eigenvectors appear in the rescaled pattern overlap vector $\tilde{\mathbf{m}}$ after the energy minimization of the LAM. Furthermore, we determined that a GL eigenvector with an eigenvalue $\lambda_k$ is activated in the pattern overlap vector under the condition $\lambda_k < \alpha + 1$ (see Methods: Analysis of the energy function of LAM). Noting that the minimum eigenvalue of GL is always zero and a smaller eigenvalue corresponds to coarser graph segmentation, this result indicates that representations of the largest community (the eigenvector with the second smallest eigenvalue, which is called the Fiedler vector) appear in LAM when $\alpha$ is slightly higher than -1. As $\alpha$ increases, eigenvectors with higher eigenvalues are also activated; thus, it is expected that the represented communities will become smaller. This analysis fits with the results shown in the previous section, especially the similarity between pattern overlaps in $\alpha \approx 1$ and Fiedler vectors (Fig 3A, 3B and 3C). Although this analysis of the energy function depends on the symmetricity of synaptic weights, we also derived the same activation thresholds of GL eigenvectors through Turing instability analysis for complex networks [37], which does not require such constraints on connectivity (see Methods: Turing instability analysis of LAM). Therefore, we can expect similar transient dynamical properties for LAM with asymmetric connections although the existence of attractors is not guaranteed in that case.

Alternatively, we can also interpret energy minimization as a combination of two conflicting optimizations. First, the minimization of $\tilde{\mathbf{m}}^{\mathrm{T}}\mathbf{L}\tilde{\mathbf{m}}$ is equivalent to the minimization of the differences between pattern overlaps $m^\mu$ for two strongly connected nodes [29]. This results in smoothing (or diffusion) on the graph, which leads to non-sparse solutions, observed as mixed representations of multiple memory patterns. Second, the term $-\tilde{\mathbf{m}}^{\mathrm{T}}\mathbf{D}\tilde{\mathbf{m}}$ is the same as the conventional Hopfield model, which leads to the activation of a single memory pattern. Minimization of the mean activity $m^0$ also helps to create sparse activity patterns. Therefore, the latter part of the energy function acts for sparsification, that is, a reduction in the number of active memory patterns. In summary, the energy function is composed of two components for

smoothness and sparsity, and the value $\alpha+1$ determines the trade-off. If $\alpha<-1$, the effect of sparsification vanishes; thus, no pattern is preferentially active. The number of active memory patterns is maximized when $\alpha$ is slightly higher than -1 because of the strong smoothing effect. As $\alpha$ increases in the region $\alpha>-1$, the number of activated patterns gradually decreases and the model approaches to the conventional Hopfield model. This intuitive interpretation also fits the actual behavior of the model.

To quantitatively validate the relationship between representations in LAM and GL eigenvectors, we performed linear regression of pattern overlaps in LAM (shown in Fig 2) using various numbers of GL eigenvectors, and calculated the variance explained (the schematic is shown in Fig 3D). Eigenvectors were chosen from those with small eigenvalues to those with large eigenvalues (when two eigenvectors were used, the smallest and the second smallest eigenvalues were chosen). The results show that the pattern overlaps in LAM with $\alpha\approx-1$ were mostly explained by small numbers of GL eigenvectors with small eigenvalues, and eigenvectors with large eigenvalues were gradually recruited as $\alpha$ increased (Fig 3E, 3F and 3G). This result is consistent with our theoretical analyses and demonstrates the mathematical mechanism behind the representations in LAM.

## Segmentation of various graph structures

So far, we have tested the model behavior using relatively simple and regular graphical structures. We then checked whether LAM could extract communities in various hierarchically organized graphs. We used the stochastic block model (SBM) [38,39] to generate random graphs with hierarchical community structures (Fig 4A). The detailed procedure is described in "Generation of random graphs" in the Methods section. We embedded the generated graph structure in LAM and obtained its attractor patterns using the same procedure and the same parameter settings as in the simulations shown in Fig 2. Depending on the value of $\alpha$, LAM extracted communities at different hierarchical levels (Fig 4B and 4C). We evaluated the explained variance of pattern overlaps by GL eigenvectors and found the same tendency as in the non-random graphs analyzed in Fig 3, where the contributions of GL eigenvectors with higher eigenvalues increased as $\alpha$ increased (Fig 4D). We quantified the similarity of the attractor patterns obtained at each hierarchical level of the communities. We calculated an average pattern correlation at each level by collecting node pairs belonging to the same community at level $h$ but to different sub-groups at level $(h+1)$ (the latter condition was not applied if level $h$ was the bottom level). As shown in Fig 4E, the average pattern correlations were high at all levels of communities when $\alpha$ was close to -1. As $\alpha$ increased, the average correlations at level 1 (the top level) dropped first, and those at levels 2 and 3 followed in this order, indicating the emergent representations of hierarchical communities that are parameter dependent. We varied the values of the parameters, that is, the number of nodes, the degree of hierarchy, the number of communities at each hierarchy level, and random seeds for sampling the structures and patterns. We observed the same tendency regardless of the settings of the random graphs (S6 and S7 Figs). These results show that LAM can generate multiscale representations of community structures embedded in random graphs in which we destroy the structural regularity while retaining the bias in connection probabilities, which defines the communities.

Based on the relationship with GL, we also tested graph-based image segmentation by LAM, which is a well-established application of GL [27]. We assigned a random binary pattern to each pixel and defined hetero-associative links between pixels based on spatial proximity and similarity of RGB values, similar to a previous study [27]. LAM (containing 30000 neurons) successfully extracted large segments corresponding to a GL eigenvector when the auto-associative weight $\alpha$ was close to $-1$ (Fig 5). When $\alpha$ was increased, the LAM extracted

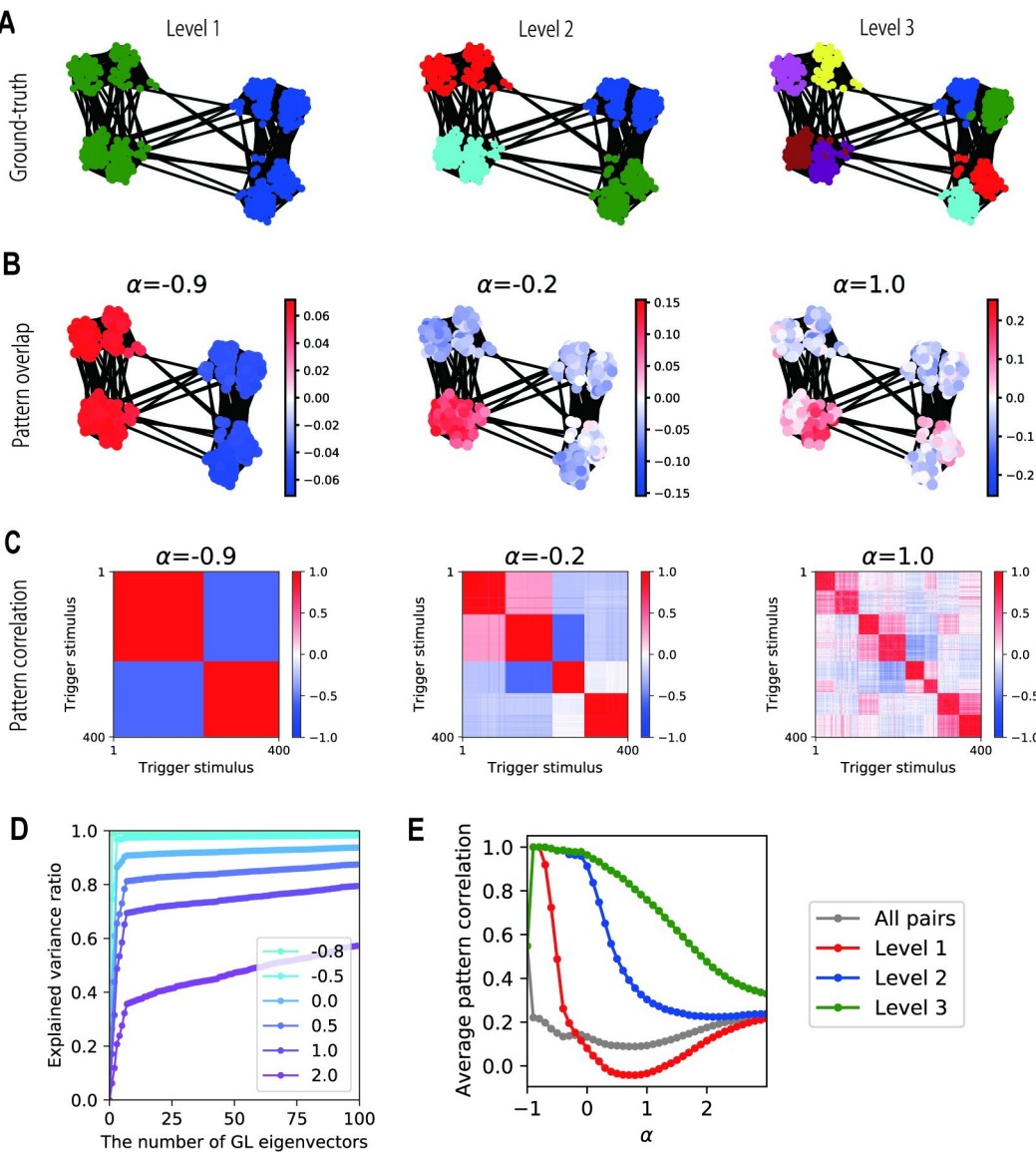

**Fig 4. Representations of LAM for a random graph with a hierarchical community structure.** (A) A structure of an example random graph (400 nodes) and three hierarchical levels of communities. (B) Pattern overlaps of attractors in LAM. (C) Pattern correlation matrices between attractors in LAM. (D) The explained variance ratio in linear regressions of pattern overlaps by various numbers of GL eigenvectors. The color indicates the value of $\alpha$. (E) Average pattern correlations between attractor patterns of node pairs in each hierarchy. A level-$h$ pair is in the same level-$h$ community and not in the same level-$(h+1)$ community if h<H.

relatively small segments (Fig 5). This result shows that LAM is also applicable to non-ideal graphs constructed from real-world data.

## Finding subgoals by graph-based representations and novelty detection

One of the important applications of GL eigenvectors is to find appropriate subgoals for hierarchical reinforcement learning [8,9]. In this framework, sets of primitive actions (options) are optimized through learning to reach the subgoals. Desirable subgoals are "bottlenecks", which are shared by many trajectories on the state-transition graph. GL eigenvectors have been used

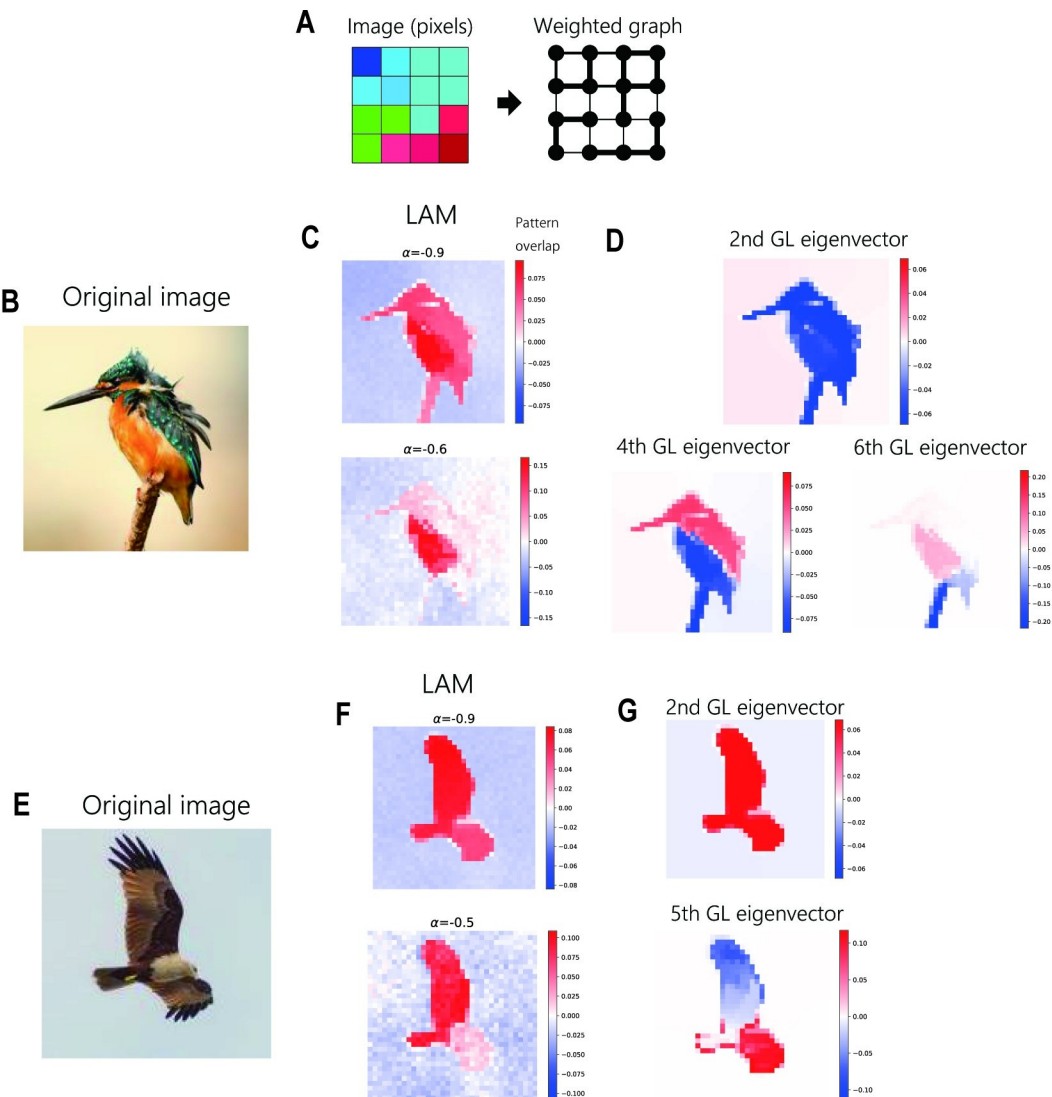

**Fig 5. Image segmentation by LAM.** (A) Conversion of images into a weighted graph. We regarded each pixel as a node and determined link weights by spatial proximity and similarity of RGB values. (B,E) Original hi-resolution images used for the segmentation task. We used down-sampled images for the construction of graphs. (C,F) Pattern overlaps obtained after the simulation of LAM with different values of $\alpha$. (D,G) Representative GL eigenvectors corresponding to segments obtained by LAM.

to identify bottlenecks through graph segmentation. We tested whether representations in LAM can also be used for subgoal finding by comparing the results between LAM and GL.

To identify the bottlenecks, we calculated the "novelty index" of each node, which measures the expected changes of representations caused by the movements from a node to surrounding nodes (for the mathematical definition, see Methods: Definition of the novelty index for subgoal finding). In hierarchical reinforcement learning, subgoals are treated as pseudo-rewards for agents. It is biologically natural to treat novelty as a pseudo-reward because dopamine cells are activated by not only reward but also novelty [40]. With GL, we constructed low-dimensional representations of nodes from GL eigenvectors with low eigenvalues (Laplacian eigenmap)[29]. On the other hand, with LAM, the activity patterns in the attractor states were directly used as representations.

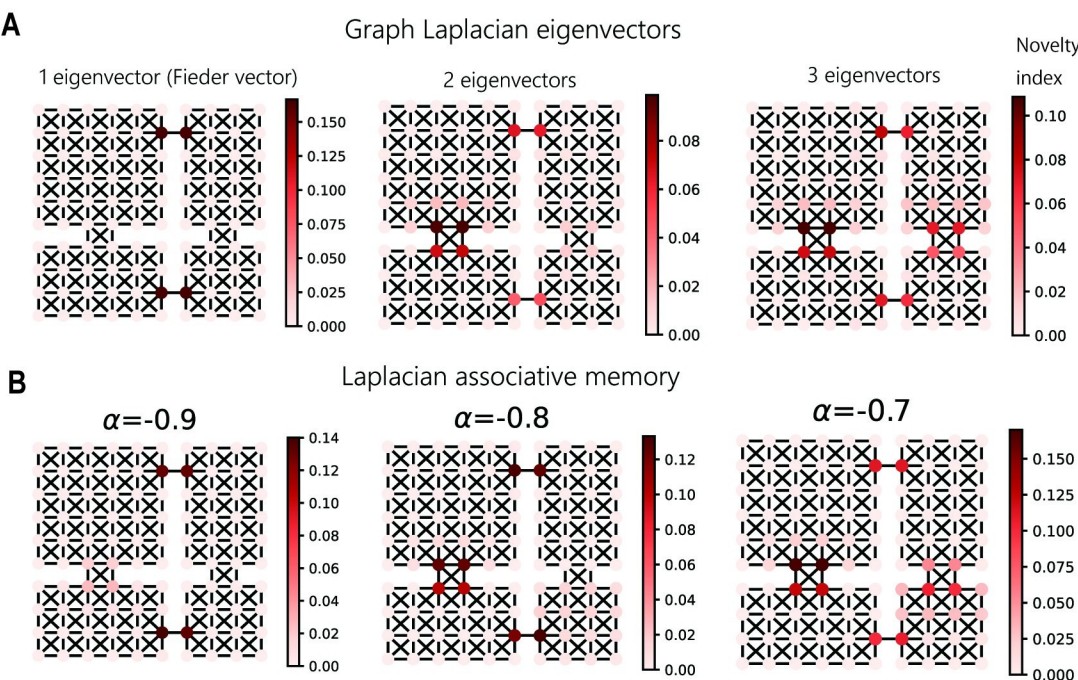

**Fig 6. Subgoal finding by novelty detection with representations in LAM.** (A) Subgoal finding using low-dimensional (1, 2, or 3) representations constructed from graph Laplacian eigenvectors. The color indicates the novelty index for each node. (B) Subgoal finding using representations in LAM obtained by different values of $\alpha$.

With GL, the novelty index successfully detected the nodes located at the bottlenecks (doorways) between distinct compartments (Fig 6A). Furthermore, the sensitivity of bottleneck detection was controlled by the dimension of representation vectors, and the use of a single eigenvector with a small eigenvalue (Fiedler vector) extracted the narrowest bottlenecks, and higher dimensional representations enabled the detection of other bottlenecks. We obtained an equally good performance with LAM (Fig 6B). In LAM, the auto-associative weight $\alpha$ regulates the number of active GL eigenvectors in representations; hence, it changes the sensitivity of bottleneck detection. This result demonstrates that novelty detection in LAM enables multi-resolution subgoal finding comparable to GL eigenvectors. The idea of using novelty has been suggested in the literature on hierarchical reinforcement learning [36] but LAM provides a more biologically plausible mechanism based on a neural network.

### Chunked sequential activities in asymmetric LAM

So far, we have analyzed attractor patterns in LAM with symmetric links. Next, we show the dynamic properties of asymmetric LAM. We constructed an asymmetric LAM with a ring-shaped graph in which link weights were slightly stronger in one direction than in the opposite direction (Fig 7A). We simulated the neural activity while continuously changing the value of the auto-associative weight $\alpha$ (Fig 7B, top). The network generated a sequential activity in which embedded memory patterns were consecutively retrieved at a variable speed (Fig 7B, bottom). Rapid state transitions occurred at specific moments when the value of $\alpha$ became negative and close to -1 (Fig 7D), at which the distribution of pattern overlaps was maximally expanded (Fig 7C). This result indicates that negative auto-associative weights in asymmetric LAM not only generate macroscopic representations for large communities but also increase the sensitivity to asymmetricity in link weights and facilitate sequential transitions across memories.

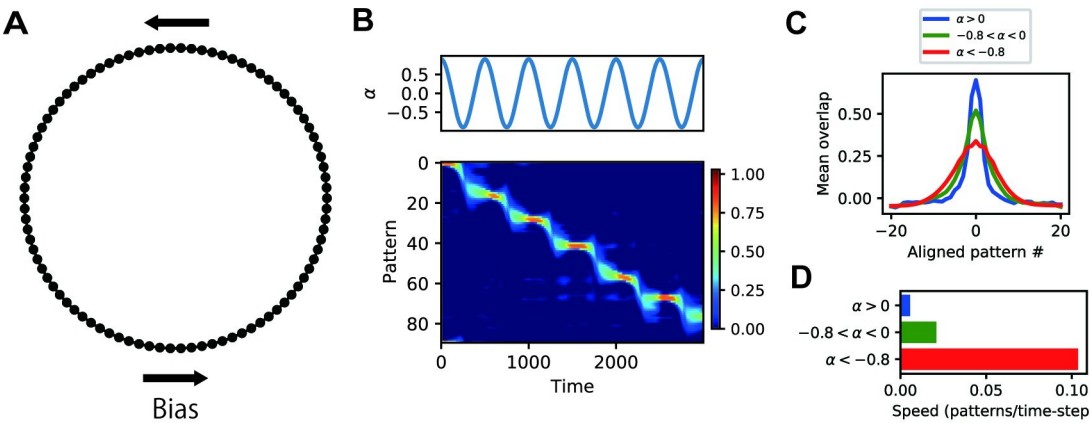

**Fig 7. Parameter-dependent sequential activities in asymmetric LAM.** (A) A ring structure of hetero-associative links for the simulation of asymmetric LAM. Hetero-associative weights were biased towards one direction. (B) The time course of $\alpha$ (top) and pattern overlaps (bottom) in the simulation of asymmetric LAM. Negative pattern overlaps were truncated to zero. (C) Peak-aligned mean pattern overlap distributions in different ranges of $\alpha$. (D) Mean speed of the peak shift of the pattern overlap distribution in different ranges of $\alpha$.

Motivated by this dynamic property and the relationship between LAM and GL, we examined whether the sequential activities in asymmetric LAM are chunked according to the communities in hetero-associative links. For the simulations, we specifically focused on the hippocampal theta sequences in which chunking was experimentally observed [32]. We assumed a virtual animal running on a ring-shaped track, and modeled the hippocampus of the animal by asymmetric LAM with a ring-shape hetero-associative link structure (Fig 8A). We simulated the neural activities of LAM with a fixed parameter $\alpha = -0.9$ while regularly stimulating a cell assembly encoding the current location of the animal. Using this procedure, we generated repeated sequential activities along the ring which resembled theta sequences (Fig 8B).

In this model, we tested three hetero-associative link structures (see S8 Fig for details of structures). In a uniform ring structure without chunks (Fig 8C), sequential activities propagated homogenously (Fig 8F and 8I). When the structure had local bottlenecks (Fig 8D), sequential propagation was constrained at the bottlenecks (Fig 8G and 8J). This result is analogous to that of the symmetric LAM with a four-room graph (Fig 2K). Finally, when the structure was chunked by local over-representations (Fig 8E) which were implemented as densely connected nodes, sequential propagation was also chunked at over-represented locations and the representations strongly correlated within chunks (Fig 8H and 8K). These effects resemble the chunking of theta sequences observed in animal experiments (Fig 8L) [32]. Although both the bottleneck model and the over-representation model exhibited chunking effects, the over-representation model was particularly consistent with the experimental observation because theta sequences were segmented at salient landmarks and rewards [32] which are over-represented by hippocampal place cells [41,42]. This result demonstrates that LAM provides a unified mechanism for graph-based representations [6,7] and chunking of sequential activities [32].

## Discussion

In this paper, we proposed Laplacian associative memory (LAM), an extension of Hopfield-type network models to compute community structures in hetero-associative links. While structural segmentation has been attempted by hierarchical networks with different time

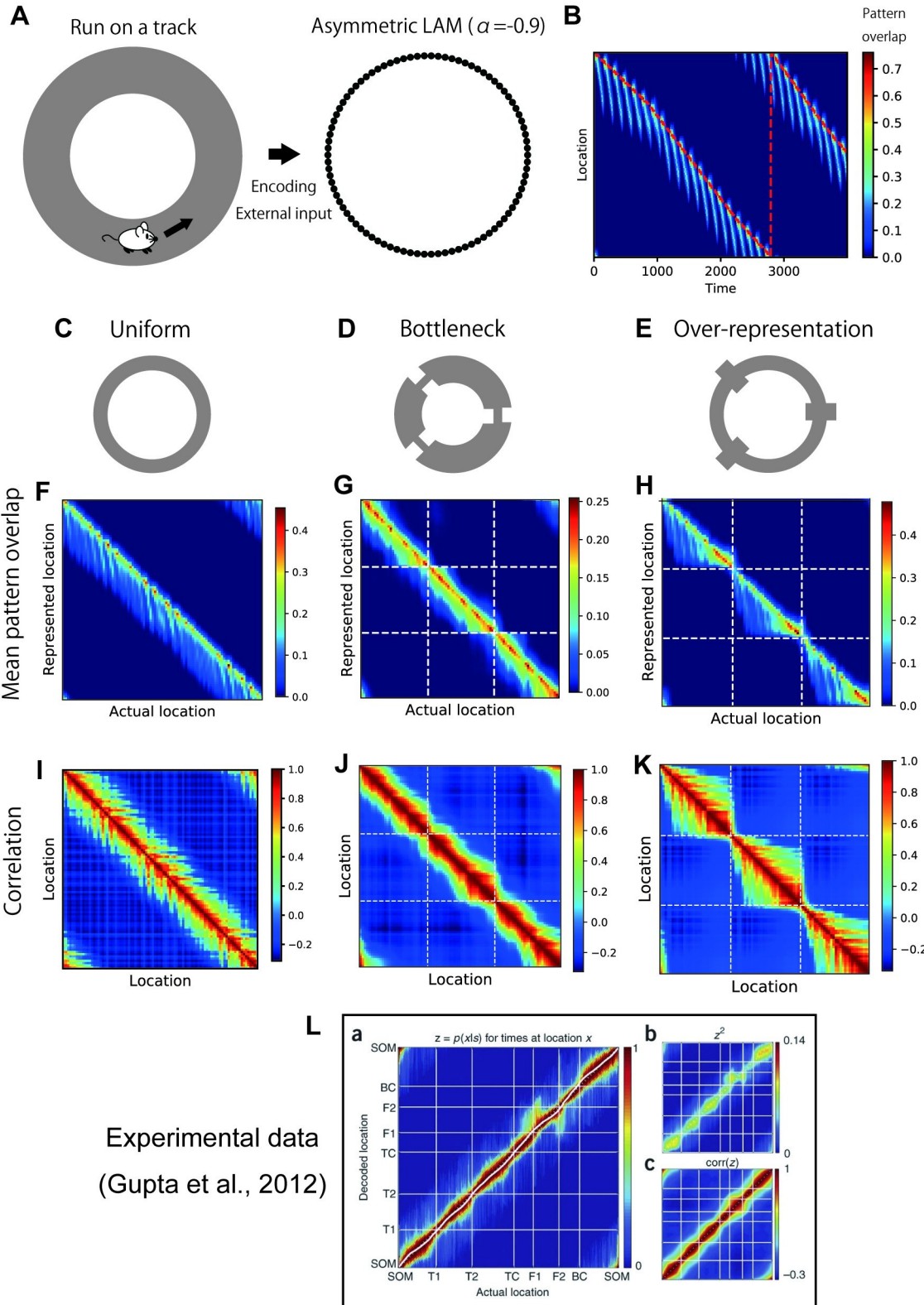

**Fig 8. Chunked sequential activities in asymmetric LAM.** (A) Schematics for the simulation setting. (B) Patten overlaps of simulated neural activities in the uniform model. The red dotted line indicates the actual location of the virtual animal. (C-E) Schematics of the hetero-associative link structure. A uniform ring (C), a ring chunked with local bottlenecks (D), and a ring

chunked with local over-representation (E). The details are shown in S8 Fig. (F-H) Mean pattern overlaps at each actual location of a virtual animal in the simulation of a uniform model (F), a bottleneck model (G), and an over-representation model (h). We truncated negative pattern overlaps to zero in these figures. (I-K) Correlations between mean pattern overlaps at different locations in the simulation of a uniform model (I), a bottleneck model (J), and an over-representation model (K). The white dotted lines in G, H, J, and K indicate chunk borders (bottlenecks or over-representations). (L) Experimental data showing segmentation of decoded spatial representations in theta sequences. The labels indicate landmarks on the track. SOM: start of maze, T1 and T2: turns, TC: top corner, F1 and F2: feeders, BC: bottom corner. The image in L was reproduced with permission from ref. [32], Springer Nature.

constants [43], our model provides a novel framework for multiscale information processing in a single network and accounts for experimentally observed graph-based representations [6,7]. Furthermore, we showed that LAM with asymmetricity can generate chunked sequential activities that reproduce experimentally observed chunking of theta sequences [32]. Notably, a model parameter crucial for segmentation can be interpreted as the strength ratio of local (assembly specific) and global (non-specific) inhibitory feedback. This interpretation offers a novel insight into the computational roles of inhomogeneous neuromodulations of interneuron circuits, which has been the focus of recent experimental studies [30,31].

We interpreted the model parameter $\alpha$ which controls the scale of the representation as a balance between global and local inhibition (Eq (3)). In light of the recent experimental evidence, we further speculate on the specific circuit mechanisms that regulate the parameter $\alpha$. In the visual cortex, somatostatin-expressing (SOM) interneurons and parvalbumin-expressing (PV) interneurons are considered to serve as anatomically global and local inhibitory feedback, respectively [44]. Furthermore, the cholinergic activation of vasoactive intestinal peptide-expressing (VIP) interneurons selectively inhibits SOM interneurons, which changes the balance between SOM and PV interneurons [30,31]. Therefore, it seems reasonable to hypothesize that acetylcholine elicits a local-inhibition-dominant (PV-dominant) state in cortical circuits and hence generates the macroscopic representations corresponding to large communities. This hypothesis is consistent with the computational model of cortical inference in which acetylcholine signals uncertainty [45] because acetylcholine creates mixed and ambiguous representations over many states. However, our model offers a novel prediction that the uncertainty of the inference (expansion of probabilistic distributions) is constrained by communities in the graph structure behind the experience. Our hypothesis also predicts that the malfunctioning of PV interneurons results in a deficit of processing uncertainty and macroscopic information, and difficulty of transition between states. This may be consistent with symptoms of schizophrenia and its likely cause [46–48]. These possibilities should be pursued by more specific and detailed modeling and fitting of the experimental data.

We used graph-based representations in LAM for the subgoal findings in HRL [8,9]. Another way to perform reinforcement learning with graph-based representations is to use a successor representation [49]. Successor representation provides a prediction of near-future state occupancy from current states, which is useful for value estimation, and has been shown to be consistent with many experimental findings in hippocampal information representations [10], including representational similarity within community structures shown in this study. The representations in our model share similar properties with successor representation as its eigendecomposition yields GL eigenvectors [10] and modifying the timescale of representation results in changing eigenvalues. Therefore, we expect a theoretical connection between the successor representation and our model. However, we note that our model directly exhibits the Fiedler vector in $\alpha \approx -1$, but the successor representation cannot provide such representation in any parameter setting (see Methods: eigenvalues of successor representations). Therefore, in the current form, there is a small quantitative difference between the representations in our model and the successor representation; hence, our model does not directly support

value estimation. However, the advantage of our model is the clear interpretation of the circuit-level mechanism in contrast that the circuit-level dynamics for computing successor representation are still unknown. Which model better explains the experimental observations and whether our network model (with some modifications) exhibits successor representation is an interesting open problem.

In asymmetric LAM, we found that negative auto-associative weights facilitate sequential transitions across memory patterns. Previously, we had found that negative auto-association significantly increased the sensitivity of correlated attractors to external perturbation [23]. We speculate that the changes in the propagation speed presented here depend on a similar mechanism. If the auto-association is strongly positive, attractors are stable and invulnerable to directional biases in link weights. However, as $\alpha$ approaches -1, the attractors are gradually instabilized and become sensitive to weight biases and external perturbations. This property suggests that macroscopic representations are dynamic in the brain, and are unlikely to serve as robust working memory as conventional attractor networks [20,24,25].

We found that both local bottlenecks and over-representations induce chunking of sequential activities in asymmetric LAM. The over-representation model is particularly interesting because it accounts for the role of salient landmarks and rewards that are over-represented by place cells [41,42]. We may be able to apply a ring-shaped structure with two over-representations for modeling the typical experiments in which animals run back and forth on a 1-D track to obtain rewards at both ends, considering that many place cells are direction-selective in such experimental settings [50]. In contrast, to the best of our knowledge, the effect of bottlenecks on hippocampal sequential activity has not been tested experimentally. An adequate design of bottlenecks does not seem to be trivial in spatial navigation tasks because animals may recognize spatial bottlenecks as salient landmarks that would be over-represented in the brain. A proper design of the task structure requires careful control of the saliency of each state.

The simple model based on the asymmetric LAM produced sequential activities similar to chunked hippocampal theta sequences [32] (Fig 8). However, hippocampal circuits generate more complex oscillatory dynamics, which are also likely to contribute to segmentation. For instance, in hippocampal replays of spatial trajectories, a boundary of chunks (a bifurcating point) in the spatial structure is locked to troughs of LFP power in concatenated sharp-wave ripples [33]. Furthermore, hippocampal circuits repeat convergence to and divergence from discrete attractors every gamma cycle during sharp-wave ripples[16]. Our simplified model cannot address the relationship between complex oscillatory dynamics and segmentation. A detailed network model involving realistic spiking neurons and inhibitory circuits is necessary to study such a relationship.

In this work, we manually tuned the values of parameters such as the number of neurons $N$, sparsity $p$, and additional inhibition $\gamma$. While the relationship between individual parameters and the performance of the conventional Hopfield models has been theoretically investigated [12,35,51,52], we have not fully understood the parameter dependence of the performance of LAM. Further theoretical analyses of the model are required to determine optimal parameter setting. Such an investigation may reveal the existence of other interesting states of the model that were not addressed in this study.

Previously, the processing of hierarchical knowledge was implemented in associative memory models by embedding artificially correlated memory patterns [53]. Such models successfully reproduced the dynamics of hierarchical information processing in the temporal visual cortex [54,55]. The relationship between our model with hetero-associative links and the previous model with correlated memory patterns is currently unclear, and worth exploring. If similar graphical computation is possible with correlated memory patterns, the brain may perform

graphical computation based not only on temporal association (hetero-associative links in our model) but also on semantic similarity between items (correlation between memory patterns). However, we emphasize that our finding that associative memory networks can autonomously compute mathematically well-defined communities in complex graphs was not known, because previous models tested only simple structures and negative auto-association was not considered.

The mechanism proposed in this paper provides a novel method for solving an arbitrary eigenvalue problem by using associative memory models. In the present model, we constructed hetero-associative weights from the normalized adjacency matrices of the graphs. However, the proposed dynamical mechanism to solve the eigenvalue problem is generic and does not depend on this specific condition. For example, if we employ a covariance matrix between encoded variables for a hetero-associative weight matrix, a network is theoretically expected to perform principal component analysis. Because eigenvalue problems ubiquitously appear in applied mathematics and machine learning, other computational methods may also be mapped to brain functions through a similar mechanism. Our model suggests much more powerful computing ability of associative memory models than previously thought and may provide a bridge between artificial intelligence and brain science.

## Methods

### Definition and mathematical properties of graph Laplacian

Let us assume a symmetric graph that has an adjacency matrix $\mathbf{A}$ whose element $A_{ij}$ denotes the existence of an edge with 0 and 1 (unweighted graphs) or the weight of the edge (weighted graphs) between node $i$ and node $j$. We also define a degree matrix $\mathbf{D}$, in which the diagonal elements are degrees (the number of edges connected to each node) $d_i = \sum_j A_{ij}$ and other elements are zero. The graph Laplacian is a matrix defined as $\mathbf{L} = \mathbf{D} - \mathbf{A}$. There are two ways of normalization: a symmetric one $\mathbf{L}^{\text{sym}} = \mathbf{D}^{-\frac{1}{2}}\mathbf{L}\mathbf{D}^{-\frac{1}{2}} = \mathbf{I} - \mathbf{D}^{-\frac{1}{2}}\mathbf{A}\mathbf{D}^{-\frac{1}{2}}$ and an asymmetric one $\mathbf{L}^{\text{asym}} = \mathbf{D}^{-1}\mathbf{L} = \mathbf{I} - \mathbf{D}^{-1}\mathbf{A}$ ($\mathbf{I}$ is an identity matrix). These two matrices have similar properties qualitatively [28].

An important characteristic of the graph Laplacian matrix is that its eigenvectors provide optimal graph segmentation. Here optimality is defined by the min-cut criterion that prefers a two-fold division of a graph obtained by cutting the minimum number of edges. It has been proven that min-cut graph segmentation can be performed by solving the generalized eigenvalue problem $\mathbf{L}\mathbf{y} = \lambda\mathbf{D}\mathbf{y}$, or equivalently, eigenvectors of normalized graph Laplacian $\mathbf{L}^{\text{sym}}$ and $\mathbf{L}^{\text{asym}}$ [27,28]. The sign of each element in the eigenvector $\mathbf{y}$ indicates a segment to which each node should be assigned, and multiple eigenvectors correspond to the two-fold segmentation at levels, depending on their eigenvalues. The eigenvector with the second smallest eigenvalue (Fiedler vector) is regarded as the best non-trivial solution which corresponds to the largest community structure (which achieves minimum cut) in the graph. Eigenvectors with larger eigenvalues are suboptimal solutions perpendicular to other eigenvectors, and tend to subdivide large communities into subclusters.

Another useful interpretation of graph Laplacian eigenvectors is low-dimensional representations of nodes in the graph which is called the Laplacian eigenmap [29]. The generalized eigenvalue problem $\mathbf{L}\mathbf{y} = \lambda\mathbf{D}\mathbf{y}$ gives perpendicular solutions for $\min_{\mathbf{y}} \mathbf{y}^{\text{T}}\mathbf{L}\mathbf{y}$ subject to $\mathbf{y}^{\text{T}}\mathbf{D}\mathbf{y} = 1$ and the eigenvalue indicates the minimized value. Because of the relationship

$$\mathbf{y}^{\text{T}}\mathbf{L}\mathbf{y} = \frac{1}{2}\sum_{i,j} A_{ij}(y_i - y_j)^2, \tag{10}$$

minimization of $\mathbf{y}^{\text{T}}\mathbf{L}\mathbf{y}$ can be regarded as assigning values $y_i$ to nodes such that strongly connected nodes are represented by close values. In this sense, low-dimensional representations

constructed from graph Laplacian eigenvectors with low eigenvalues capture the graph structure through their similarity, which is an appropriate property for nonlinear dimensionality reduction.

## Analysis of the energy function of LAM

Here we consider a symmetric normalization model in which hetero-associative links are constructed as $\mathbf{H} = \mathbf{D}^{-\frac{1}{2}}\mathbf{A}\mathbf{D}^{-\frac{1}{2}}$. As in the main text, the energy function is

$$E = -\frac{1}{NV}\sum_{i=1}^{N}\sum_{j=1}^{N} w_{ij}x_i x_j, \tag{11}$$

We define a pattern overlap

$$m^\mu = \frac{1}{NV}\sum_{i=1}^{N} \tilde{\xi}_i^\mu x_i, \tag{12}$$

and mean activity $m^0 = N^{-1}\sum_{i=1}^{N} x_i$. By substituting $w_{ij}$ defined in Eq (2), we can rewrite the energy function using the pattern overlaps as

$$E = -\sum_{\mu=1}^{P}\sum_{\nu=1}^{P} H_{\mu\nu}m^\mu m^\nu - \alpha\sum_{\mu=1}^{P}(m^\mu)^2 + (\alpha+1)\gamma V^{-1}(m^0)^2. \tag{13}$$

By using the pattern overlap vector $\mathbf{m} = (m^1,\ldots,m^P)^{\mathrm{T}}$ and symmetric normalized graph Laplacian $\mathbf{L}^{\mathrm{sym}} = \mathbf{I}-\mathbf{H}$, the energy function can be written in vector form:

$$E = \mathbf{m}^{\mathrm{T}}\mathbf{L}^{\mathrm{sym}}\mathbf{m} + (\alpha+1)[-\mathbf{m}^{\mathrm{T}}\mathbf{m} + \gamma V^{-1}(m^0)^2]. \tag{14}$$

By rescaling $\mathbf{m}$ by the degree matrix $\mathbf{D}$ as $\tilde{\mathbf{m}} = \mathbf{D}^{-\frac{1}{2}}\mathbf{m}$, we further obtain

$$E = \tilde{\mathbf{m}}^{\mathrm{T}}\mathbf{L}\tilde{\mathbf{m}} + (\alpha+1)[-\tilde{\mathbf{m}}^{\mathrm{T}}\mathbf{D}\tilde{\mathbf{m}} + \gamma V^{-1}(m^0)^2], \tag{15}$$

where $\mathbf{L}$ is unnormalized graph Laplacian.

To see the relationship with graph Laplacian eigenvectors more quantitatively, we expand the overlap vector $\mathbf{m}$ by a linear combination of eigenvectors of the symmetric normalized graph Laplacian $\boldsymbol{\phi}_k$ (with corresponding eigenvalues $\lambda_k$) as

$$\mathbf{m} = \sum_{k=1}^{P} c_k\boldsymbol{\phi}_k. \tag{16}$$

Then, the energy function can be written as

$$E = -\sum_{k=1}^{P}(c_k)^2[\alpha+1-\lambda_k] + \gamma V^{-1}(\alpha+1)(m^0)^2. \tag{17}$$

If $\gamma = 0$, the minimization of this energy requires $c_k \neq 0$ if $\lambda_k < \alpha+1$, which gives the approximate threshold for the activation of an eigenvector $\boldsymbol{\phi}_k$ in the representation (note that the actual threshold can be shifted because of $\gamma > 0$).

## Turing instability analysis of LAM

For the analysis, we first replaced the step function $\Theta(x)$ in Eq (1) by a differentiable monotonically increasing function $f(\epsilon x)$ that converges to $\Theta(x)$ in the limit of $\epsilon \to \infty$ (e.g. a logistic

function $f(\epsilon x) = (1+\exp(-\epsilon x))^{-1}$. As in the main text, we define the pattern overlap as

$$m^\mu = \frac{1}{NV} \sum_{i=1}^{N} \tilde{\xi}_i^\mu x_i. \tag{18}$$

From Eqs (1)(2), the dynamics of overlaps can be obtained as

$$\dot{m}^\rho = -m^\rho + \frac{1}{NV} \sum_{i=1}^{N} \tilde{\xi}_i^\rho f\left( \epsilon V^{-1} \sum_{\mu=1}^{P} \sum_{\nu=1}^{P} (\alpha \delta_{\mu\nu} + H_{\mu\nu}) \tilde{\xi}_i^\mu m^\nu \right) \tag{19}$$

Next, with vectors $\mathbf{m} = (m^1,\ldots,m^P)^{\mathrm{T}}$, $\tilde{\boldsymbol{\xi}}_i = (\tilde{\xi}_i^1,\ldots,\tilde{\xi}_i^P)^{\mathrm{T}}$, and the hetero-associative weight matrix $\mathbf{H} = (H_{\mu\nu})_{1\leq\mu\leq P, 1\leq\nu\leq P}$, we obtain a vector representation as

$$\dot{\mathbf{m}} = -\mathbf{m} + \frac{1}{NV} \sum_{i=1}^{N} \tilde{\boldsymbol{\xi}}_i f(\epsilon V^{-1} \tilde{\boldsymbol{\xi}}_i^T (\alpha \mathbf{I} + \mathbf{H})\mathbf{m}) \tag{20}$$

where $\mathbf{I}$ denotes the identity matrix. When $N$ and $P$ are sufficiently large and the memory patterns are random, $\mathbf{m} = \mathbf{0}$ is an equilibrium point for this dynamical equation. Furthermore, in this condition, the matrix $\frac{1}{NV} \sum_{i=1}^{N} \tilde{\boldsymbol{\xi}}_i \tilde{\boldsymbol{\xi}}_i^{\mathrm{T}}$ becomes a correlation matrix for random memory patterns, which can be approximated by an identity matrix. Therefore, we obtain the following equation by linearizing $f(x)$ around $\mathbf{m} = \mathbf{0}$:

$$\dot{\mathbf{m}} = -\mathbf{m} + \epsilon V^{-1} f'(0)[(\alpha+1)\mathbf{I} - \mathbf{L}]\mathbf{m} \tag{21}$$

Here, we defined $\mathbf{L} = \mathbf{I} - \mathbf{H}$ (either symmetric or asymmetric normalized graph Laplacian). Finally, we expand $\mathbf{m}$ with eigenvectors $\boldsymbol{\phi}_n$ ($n = 1,\cdots,P$) of matrix $\mathbf{L}$ as follows

$$\boldsymbol{m} = \sum_{n=1}^{P} d_n \exp(\beta_n t) \boldsymbol{\phi}_n \tag{22}$$

Substituting this into the linearized equation yields

$$\sum_{n=1}^{P} [(\alpha + 1 - \lambda_n)\epsilon V^{-1} f'(0) - 1 - \beta_n] d_n \exp(\beta_n t) \boldsymbol{\phi}_n = \mathbf{0} \tag{23}$$

where $\lambda_n$ is the eigenvalue of $\boldsymbol{\phi}_n$. This equation has non-trivial solutions ($d_n \neq 0$) only if $\beta_n = (\alpha+1-\lambda_n)\epsilon V^{-1} f'(0)-1$, which gives exponential growth rates along each eigenvector around $\mathbf{m} = \mathbf{0}$. If there exists a positive growth rate, the network becomes unstable along the corresponding eigenvectors; otherwise the network is stabilized at $\mathbf{m} = \mathbf{0}$. In the limit of $\epsilon \to \infty$ ($f(\epsilon x) \to \Theta(x)$), the sign of $\beta_n$ is solely determined by the sign of $\alpha+1-\lambda_n$. This result suggests that the overlap vector $\mathbf{m}$ is activated (instabilized) along the $k$-th eigenvector of the graph Laplacian matrix $\mathbf{L}$ if $\alpha > \lambda_k - 1$ ($\lambda_k$ is the eigenvalue for the $k$-th eigenvector).

## Simulations of the network model

In the numerical simulations, we used the decomposed asymmetric normalization model defined by Eq (3) unless specified otherwise. However, we decomposed only the global inhibition terms because the decomposition of local inhibition did not alter the model behavior. Therefore, we did not specify the value of $\alpha_{\max}$ in the simulations. We first initialized activities using one of the memory patterns ($x_i[0] = \xi_i^\mu$) and updated activities using a discretized

version of Eq (1):

$$x_i[t+1] = x_i[t] + \eta\left[-x_i[t] + \Theta\left(\sum_{j=1}^{n} w_{ij}x_j[t] + e_i[t]\right)\right], \tag{24}$$

where $e_i[t]$ is an external input applied in simulations in Fig 8. We used $\eta = 0.01$ for simulations with symmetric graphs (Figs 2–6) and $\eta = 0.1$ for the simulations of the sequential activities (Figs 7 and 8). The number of neurons was $N = 30000$ for image segmentation tasks, and $N = 10000$ for all other simulations. The additional inhibition parameter was $\gamma = 0.6$ for image segmentation and simulations of sequential activities, and $\gamma = 0.3$ for all the other simulations. Sparsity $p$ was 0.1 (approximately 10% of neurons are active in each pattern) throughout the study unless specified otherwise.

Attractor patterns of the network model with symmetric graphs were obtained by simulating 3,000 time-steps. The simulations of sequential activities in Fig 7 was performed for 10,000 time-steps. Simulations of sequential activities in Fig 8 were performed for 30,000 time-steps and repeated three times using different random seeds for each setting. We averaged the mean pattern overlaps at each location and the correlations between mean pattern overlaps over those three trials. We truncated the negative mean pattern overlaps to zero in this calculation.

We counted the number of active patterns by counting the number of $\mu$ that satisfies both conditions $m^\mu > 0.05$ and $m^\mu > \frac{1}{2}\max_{\mu'} m^{\mu'}$.

## Settings for image segmentation

For image segmentation tasks, we took images from pxhere (https://pxhere.com/). We trimmed and down-sampled the images so that they contained 1000–1500 pixels (e.g. $P \approx 1000$). We note that the images shown in the figures are the ones before down-sampling. We constructed link weights by the same way with Shi & Marik (2000) [27]:

$$A_{ij} = \begin{cases} \exp\left(-\dfrac{\|\mathbf{F}_i - \mathbf{F}_j\|_2^2}{\sigma_\mathrm{I}} - \dfrac{\|\mathbf{X}_i - \mathbf{X}_j\|_2^2}{\sigma_\mathrm{X}}\right), & \text{if } \|\mathbf{X}_i - \mathbf{X}_j\|_2 < r \\ 0 & , \quad \text{otherwise} \end{cases} \tag{25}$$

where vectors $\mathbf{F}_i$ and $\mathbf{X}_i$ denote the RGB value (normalized between 0 and 1) and the spatial location of pixel $i$, respectively. The parameters were $\sigma_\mathrm{I} = 0.1$, $\sigma_\mathrm{X} = 4$, $r = 5$. After setting the values, we performed asymmetric normalization of the weights to obtain hetero-associative weights.

## Generation of random graphs

We used the stochastic block model (SBM) [38,39] to generate random graphs with community structures. In SBM, nodes are separated into several groups and the connection probabilities within and between groups are given by a matrix. For example, a connection probability matrix for a graph with three groups can be expressed as

$$\begin{pmatrix} 0.9 & 0.1 & 0.05 \\ 0.1 & 0.9 & 0.05 \\ 0.05 & 0.05 & 0.9 \end{pmatrix}. \tag{26}$$

This matrix indicates that the connection probability within the same group is 0.9, the connection probability between groups 1 and 2 is 0.1, and the connection probability is 0.05 for the other combinations.

Hierarchical community structures were generated as follows. We set the following parameters: the number of nodes $P$, number of hierarchies $H$, number of divisions at each hierarchy $D$, baseline connection probability $q$, and probability ratio $\epsilon$ ($\epsilon < 1$). At the root of the hierarchy (level 0), all of $P$ nodes were assigned to a single group. At each of the lower levels, we hierarchically divided each group into $D$ subgroups (assignment of each node was uniformly random). Accordingly, there were $D^H$ groups in the bottom level of hierarchy (level H), and we defined SBM for these bottom-level groups. The connection probability within the same bottom-level group was $q$, and connection probabilities between groups in the same upper-level group were decreased proportionally as $\epsilon^{H-h}q$ for level $h$. For example, in the setting $H = D = 2$, there were four groups at the bottom level and the connection probability matrix between them was given as

$$
\begin{pmatrix}
q & \epsilon q & \epsilon^2 q & \epsilon^2 q \\
\epsilon q & q & \epsilon^2 q & \epsilon^2 q \\
\epsilon^2 q & \epsilon^2 q & q & \epsilon q \\
\epsilon^2 q & \epsilon^2 q & \epsilon q & q
\end{pmatrix}.
\tag{27}
$$

In our simulations, we varied the parameters $P, H, D$ while we fixed two parameters: the average degree $c = 25$ and the probability ratio $\epsilon = 0.1$. We chose these values based on the theoretical detectability of communities in a simple setting [39]. We derived $q$ from the other parameters as follows. The expected number of connections from a node within the same bottom-level community is

$$
\frac{P}{D^H}q.
\tag{28}
$$

The expected number of connections additionally generated at the hierarchical level $h$ is

$$
\frac{(D^{H-h} - D^{H-h-1})P}{D^H}\epsilon^{H-h}q.
\tag{29}
$$

Therefore, the average degree $c$ satisfies

$$
c = \frac{P}{D^H}q + \sum_{h=0}^{H-1}\frac{D^{H-h-1}(D-1)P}{D^H}\epsilon^{H-h}\,q.
\tag{30}
$$

This yields the setting of $q$ in our simulation

$$
q = \frac{cD^H}{P}\left[1 + (D-1)\sum_{h=0}^{H-1}D^{H-h-1}\epsilon^{H-h}\right]^{-1}.
\tag{31}
$$

### Definition of the novelty index for subgoal finding

When we used GL eigenvectors, we constructed low-dimensional embedding of nodes from GL eigenvectors with low eigenvalues (Laplacian eigenmap) [29]. We regarded these low-dimensional vectors as representations of the nodes. We defined similarity between two nodes $s(\mu, v)$ by cosine similarity between the two representation vectors. The novelty index of node $\mu$ is defined as

$$
NI(\mu) = \frac{1}{2}\sum_{v}T_{\mu \to v}(1 - s(\mu, v)),
\tag{32}
$$

where $T_{\mu \to v}$ denotes the transition probability from $\mu$ to $v$ in a random walk on the graph (which is equivalent to an element in $\mathbf{D}^{-1}\mathbf{A}$). The novelty index $NI(\mu)$ spans from 0 to 1 and

indicates the average expected change of information representations that an agent experiences in a transition from node $\mu$. When we use LAM instead of GL eigenvectors, individual nodes are represented as attractor patterns triggered by memory patterns corresponding to the nodes, and similarity $s(\mu, \nu)$ is the correlation between the two attractor patterns.

## Asymmetric Laplacian associative memory and the model of a virtual animal

To construct asymmetric hetero-associative weights, symmetric graphs were converted into mutually connected asymmetric weighted graphs. We set the weight of the asymmetric links in the biased direction (including diagonal connections) to 110, and the weight in opposite directions to 90. The weight of links horizontal to the biased direction (radial connections) was 100. After constructing the adjacency matrices, we performed asymmetric normalization as in symmetric graphs.

In the simulation in Fig 8, we represented the current location of the virtual animal on the track by a continuous value $z[t]$ ranging from 0 to 90 which corresponds to 90 nodes in the uniform ring-shape graph (uniform model). The velocity $z[t+1]−z[t]$ was constant; however, we resampled the velocity from a range [0.02, 0.04] at the timings determined by the Poisson process (the mean interval was 1000 time-steps). We determined the index of stimulated pattern by truncating $z[t]$ to an integer. We stimulated the network every 150 time-steps (the uniform model and the over-representation model) or 200 time-steps (the bottleneck model). The amplitude and the length of stimulation were 0.3 and 50 time-steps, respectively.

In the bottleneck and over-representation models, we connected additional nodes at the side of the uniform ring-shape graph (as shown in S8 Fig). We did not stimulate patterns corresponding to additional nodes. We calculated the pattern overlap for each location by averaging the nodes in the central ring and additional nodes at the same location.

## Eigenvalues of successor representation

Successor representation is defined for a pair of states $s$ and $s'$ as

$$M(s, s') = \mathrm{E}\left[\sum_{t=0}^{\infty} \gamma^t \mathrm{I}(s_t = s')|s_0 = s\right], \tag{33}$$

where $\gamma$ is the discount factor. We consider a matrix of successor representation $\mathbf{M}$, transition probability matrix $\mathbf{T}$, and $\mathbf{L}^{\mathrm{asym}} = \mathbf{I}−\mathbf{T}$ is an asymmetric normalized graph Laplacian. The eigenvectors and eigenvalues of $\mathbf{L}^{\mathrm{asym}}$ are defined as $\boldsymbol{\phi}_i$ and $\lambda_i^{\mathrm{L}}$, respectively. Then, they satisfy

$$\mathbf{L}^{\mathrm{asym}}\boldsymbol{\phi}_i = \lambda_i^{\mathrm{L}}\boldsymbol{\phi}_i. \tag{34}$$

Using the relationship $\mathbf{M} = (\mathbf{I}−\gamma\mathbf{T})^{-1}$ [10,49], we can rewrite this equation with a successor representation matrix:

$$\mathbf{M}\boldsymbol{\phi}_i = (1 + \gamma(\lambda_i^{\mathrm{L}} − 1))^{-1}\boldsymbol{\phi}_i. \tag{35}$$

Therefore, the eigenvectors of the successor representation matrix are equivalent to those of the graph Laplacian, and the eigenvalues are $\lambda_i^{\mathrm{M}} = (1 + \gamma(\lambda_i^{\mathrm{L}} − 1))^{-1}$. This relationship becomes $\lambda_i^{\mathrm{M}} = (\lambda_i^{\mathrm{L}})^{-1}$ in the limit of $\gamma{\to}1$, and $\lambda_i^{\mathrm{M}} = 1$ in the limit of $\gamma{\to}0$. Therefore, although the contribution of the Fiedler vector increases as $\gamma$ goes to 1, it is impossible to have $\lambda_i^{\mathrm{M}} > 0$ for only the Fiedler vector if the graph has a sufficiently complex structure and $\lambda_i^{\mathrm{L}}$ is continuously distributed.

## Supporting information

**S1 Fig. Time evolutions of pattern overlaps in the simulation setting in Fig 2.**
(TIF)

**S2 Fig. Simulations of LAM with different numbers of neurons.** Pattern overlaps of example attractors (left) and pattern correlation matrices (right).
(TIF)

**S3 Fig. Simulations of LAM with different sparsity parameter $p$.** Pattern overlaps of example attractors (left) and pattern correlation matrices (right).
(TIF)

**S4 Fig. LAM (symmetric normalization model) extracts multi-scale representations for community structures.** (A) Pattern overlaps of example attractor patterns. (B) Correlation matrices between activity patterns in the attractor states reached from different trigger stimuli (nodes). (C) Maximum pattern overlaps obtained by various values of $\alpha$. (D) Numbers of active patterns obtained by various values of $\alpha$. (E) The ratio of variance of overlap distributions explained by various number of graph Laplacian eigenvectors. The color indicates the value of $\alpha$. We note that, in C-E, we averaged values from all attractors reached from different trigger stimuli. (F-J) Results for Karate-club network. (K-O) Results for compartmentalized rooms.
(TIF)

**S5 Fig. The change of the energy function in the simulation of symmetric normalization model.** (A) The graph used in Schapiro et al. (2013). (B)Karate club network. (C) The four-room graph.
(TIF)

**S6 Fig. Explained variance of pattern overlaps by GL eigenvectors for random graphs with hierarchical community structures.** $P$, $H$, $D$ are the number of nodes, the number of hierarchy, the number of division in each hierarchy, respectively, Two plots in each setting show results from two different random seeds (different link structures and different memory patterns).
(TIF)

**S7 Fig. Average correlations between attractors of node pairs in each hierarchy of random graphs with hierarchical community structures.** $P$, $H$, $D$ are the number of nodes, the number of hierarchy, the number of division in each hierarchy, respectively, Two plots in each setting show results from two different random seeds (different link structures and memory patterns). A level-$h$ pair is in a same community in level $h$ and not in a same community in level $(h+1)$ (the latter condition was not applied if h = H).
(TIF)

**S8 Fig. Chunked structures of hetero-associative links used for asymmetric LAM.**
(TIF)

## Acknowledgments

We are grateful to H. Shiwaku for fruitful discussion, T. Burns, N. Hiratani and T. Kurikawa for helpful comments on the manuscript. We would like to thank Editage (www.editage.com) for English language editing. We also thank OIST Scientific Computation and Data Analysis section for the technical support for scientific computing.

## Author Contributions

**Conceptualization:** Tatsuya Haga, Tomoki Fukai.

**Formal analysis:** Tatsuya Haga.

**Funding acquisition:** Tatsuya Haga, Tomoki Fukai.

**Investigation:** Tatsuya Haga.

**Methodology:** Tatsuya Haga.

**Project administration:** Tatsuya Haga.

**Software:** Tatsuya Haga.

**Supervision:** Tomoki Fukai.

**Visualization:** Tatsuya Haga.

**Writing – original draft:** Tatsuya Haga.

**Writing – review & editing:** Tatsuya Haga, Tomoki Fukai.

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
