## [Decision Letter · Decision Letter 0]

3 May 2021

Dear Dr. Haga,

Thank you very much for submitting your manuscript "Multiscale representations of community structures in attractor neural networks" for consideration at PLOS Computational Biology. As with all papers reviewed by the journal, your manuscript was reviewed by members of the editorial board and by several independent reviewers. The reviewers appreciated the attention to an important topic. Based on the reviews, we are likely to accept this manuscript for publication, providing that you modify the manuscript according to the review recommendations.

The reviewers are overall enthusiastic about your manuscript, and have a number of constructive comments. I will just add a few of my own comments:

- Please check the English grammar throughout, and ideally have a native English speaker proofread the paper.

- R3 (comments 4 and 5) asks to do statistical tests and compare to real neural data. I think that is unnecessary for this kind of paper, although you are welcome to do that if you want.

- R2 asks that you conduct additional analyses of more complex graphs. I just want to clarify the scope of these new analyses. It would be sufficient to select one random graph model and vary the number of nodes, number of communities, and maybe 1-3 levels of hierarchy.

Sincerely,

Samuel J. Gershman

Deputy Editor

PLOS Computational Biology

[LINK]

Reviewer's Responses to Questions

**Comments to the Authors:**

Reviewer #1: This work by authors Tatsuya Haga and Tomoki Fukai is both interesting and elegant. The work studies a generalized Hopfield network (called LAM), capable of generating segmented representations of the stored memory patterns. The authors show the link between LAM and graph-theoretic methods. I would recommend the publication of this work, when some minor points are clarified (listed below).

1. Hebbian learning is mentioned, and some derivations are made in the methods, however is it actually used in the simulations? If it is not used, am I correct to say then that H is hardcoded? Did you verify using simulations whether Hebbian learning as written in equation 8 would lead to the correct weight structure when doing a random walk on the graph?

2. I have some trouble understanding going from equation 10 on line 457 to the equation on line 460. Could you provide some more explanation?

3. How do you choose alpha_max? Is this a constant you set for each type of graph or the result of some derivation?

4. It might be helpful for the reader to show pattern overlaps as a function of simulated time steps for the graphs in figure 2 (for example in a suppl figure?). Essentially like you also did figure 5B. This is a suggestion only of course.

5. Line 550-551: only one of those conditions should be valid? What does it mean when the overlap is 0.05.

6. Line 126-127 is confusing to me. In the first half of the sentence you write that the weights are non-negative but in the second part of the sentence the range is from -1 (negative) to alpha_max. Also, what is the meaning of alpha smaller than -1 in fig2?

7. If I understand it correctly, this novelty index is a way to set the adjacency matrix of the graph and as such set H? Could you elaborate on s(mu,nu) used in the methods in equation 30?

Reviewer #2: Summary:

In ``Multiscale representations of community structures in attractor neural networks,'' the authors study the brain's ability to represent hierarchical structure through the use of Laplacian associative memory (LAM). The authors study three commonly used graphs: a 4-regular graph used in graph learning experiments, a Karate-club network, and a compartmentalized room network. First, the authors introduce LAMs, which are a generalized form of Hopfield networks with hetero-associative weights. The authors then use the the three commonly used graph types as the hetero-associative weights, and demonstrate that the strength of auto-association (determined by $\\alpha$) tunes the representational overlap of LAM attractors. Next, the authors rewrite the Hopfield energy function as a function of two competing terms: $\\tilde{m}^T L \\tilde{m}$ and $\\tilde{m}^T D \\tilde{m}$ and demonstrate that LAM dynamics balance an optimization between the conventional Hopfield cost, $\\tilde{m}^T D \\tilde{m}$, and the pattern overlap cost, $\\tilde{m}^T L \\tilde{m}$. The authors find that the overlapped pattern (as determined by $\\alpha$) is well explained by the eigenvectors of the smallest eigenvalues of the graph Laplacian. Finally, the authors generalize the utility of their findings by using LAMs to detect graph bottlenecks, perform image segmentation, and chunk sequential activity in time for asymmetric LAMs.

Overall, I find this work to be quite interesting and impactful. While the role of graph Laplacians in detecting community structure is well studied, there is increasing interest in the neural mechanisms that underlie our mental representations of community structure. This work provides an elegant and intuitive candidate mechanism using an extension of the very well-known Hopfield model. However, there are several major concerns I have about methodology and presentation that would need to be addressed before recommending acceptance.

Comments:

- Generalizability of results to parameters: My first main concern is how generalizable these findings are to different choices of simulation parameters. I appreciate the authors' analytical relationship between the LAM energy function and graph Laplacian. However, the numerical validation of the theory, as far as I can tell, consist of a few simulations in a very specific parameter regime. I would like to see how robustly the results of Figures 2 and 3 behave under different numbers of neurons (N) and representational sparsities (p).

- Generalizability of results to other hierarchical random graphs: My second main concern is how generalizable these findings are to more complex hierarchical structure. While I very much appreciate the authors' selection of relevant graph types, I would like to see reproductions of the results of Figures 2 and 3 on other common models of hierarchical graphs. My motivation for this question is that in Figures 2 and 3, the authors only use 3 specific instances of graphs, where two of them (4-regular graph and compartmentalized room) have very regular connectivities and simple hierarchies. In contrast, real neural systems have to identify hierarchical structure in a distribution of graphs that are often irregular. Hence, I would like to see Figures 2 and 3 replicated on a family of hierarchical random graphs where you can change the number of nodes and communities, and measure 1) the effect of graph size (as measured by number of nodes), 2) the effect of randomness in graph connectivity (due to randomness in different instantiations of the hierarchical random graphs), and 3) the effect of number of communities and levels of hierarchy. Of particular importance is #3, because the three graphs that the authors currently study do not have many levels of hierarchy. The 4-regular graph and compartmentalized room structure both only have one level of hierarchy (interconnected modules), and the karate-club network is quite small.

- Presentation of methods: I found it quite difficult to read through the results, because many modeling choices and expressions were not described. Upon reading the methods, many of my questions were answered. I highly recommend that the authors reference the specific section of the methods by name that are relevant in the results. Some examples are:

= The derivation of Eq. 2, and subsequent definitions of terms such as $\\tilde{\\xi}$ and $V = p(1-p)$,

= The activity-dependent learning rule and its convergent form

= The decomposition of $w_{ij}$ into excitatory and inhibitory synaptic weights

= The pattern overlap vector in rewriting the energy function

This list is not exhaustive, but are the main parts of the paper that I found most difficult to understand. For example, I recommend the authors state something to the effect of ``(for more details about the derivation of Eq. 3, see methods: Decomposition of excitatory and inhibitory synaptic weights)'' after equation 3.

- Methodological details: I would like to see the number of neurons and sparsity ratio moved from the methods to the results. As I was reading, I would have found it intuitive and helpful for this information to be present early on in the results.

- Proof of monotonic energy decrease: The authors demonstrate in Figure S2 that the energy decreases over time. While the claim that equations 5 and 6 serve as energy functions for the LAMs are fine in the context of Figure S2, is this generally true? It is a very strong statement to say that Eq. 5 is truly and globally an energy function for the LAM. Is this statement mathematically provable?

- Rigor of notation and derivations: Several times in the text, I noticed the usage of equal signs when approximation signs should have been used. This is seen in Figure 3D where the pattern overlap is an approximate sum of the 2nd, 3rd, and 4th GL eigenvectors, but an equality is used. Another example is the statement in 192: ``As in the conventional Hopfield model, dynamics of LAM monotonically decrease this energy.'' It is unclear whether this statement is generally true, or only true for these particular examples. In general, I believe the results and methods would benefit from more care in explicitly stating the conditions for which they are true.

- Clarity of methodological derivation: While I appreciate the derivations in the results and methods, they are quite difficult to follow. I would like the authors to expand upon the derivations in the methods so that they can be more easily followed. Some specific sections that I would like to see extra explanations for are:

= Construction of hetero-associative weights. In particular, there are a lot of operations that are performed between equations 8-11. Could the authors break up the derivation into more intermediary parts, and describe using words exactly what they did (for example, exactly which variables were substituted)?

= Decomposition of excitatory and inhibitory synaptic weights. In particular, equations 12-15 are simply presented, without any explanation for why they are defined that way.

= Analysis of the energy function of LAM. Could the authors write a few sentences about what a ``pattern overlap vector'' is? I understand how it is defined in equation 17, but it would help to understand, for example, why the authors call it a ``pattern overlap vector.''

- Parameter choices: Why do the authors change the number of neurons (N=30,000 for image segmentation, N=10,000 for the rest), the time step ($\\eta = 0.01$ for symmetric simulations, and $\\eta = 0.1$ for sequences), and $\\gamma = 0.6$ for the image segmentation, and $\\gamma = 0.3$ for the rest? Is it possible to find a single parameter regime where all behaviors are observed? While it is not crucial that the authors can find such a regime, it would improve confidence in the generalizability of the results.

Reviewer #3: LAM Review

in this paper, the authors propose the Laplacian associative memory (LAM) model, an extension of Hopfield networks which additionally allows associations between different memory patterns (hetero-associations). They show empirically that this allows LAM to discover clusters of associated memories (community structure) at different scales, which has been previously demonstrated in humans and animals in behavioral and neural studies. Relatedly, they also show that LAM can also detect subgoals as bottlenecks between clusters. They also demonstrate chunked sequential activation patterns similar to hippocampal theta sequences. Finally, the authors draw explicit links between the graph Laplacian and LAM. Specifically, they show theoretically that, under certain assumptions, the energy function minimized LAM is equivalent to the objective function of graph segmentation, what can be optimized by taking the eigenvectors of the graph Laplacian. It is also noteworthy that they derive a biologically plausible implementation of LAM, which allows them to trace some of the effects to the parameter alpha which controls the balance between global and local inhibition. It’s especially cool how this balance controls the scale of the discovered clusters.

How the brain organizes the world hierarchically is a central question in neuroscience. This paper brings us closer to an answer by showing how graph clustering could be implemented in a biologically plausible circuit and confirming that it exhibits activation patterns similar to those found in neural data. The paper is written clearly and is relatively straightforward to follow, and provides a pleasant mix of theoretical analyses and simulations of neural and behavioral data. I think the paper could be published in its current form, so I’m going to give some relatively minor suggestions.

1. It’s a little unclear to me whether LAM is really conceptually different from associative models for learning temporal sequences. Specifically, one way to learn the structure of the graph is by doing a random walk (or some other kind of traversal) and establishing connections between successive stimuli. Indeed, this is how humans in the Schapiro et al. task learn the graph, and also how the RNN they propose in one of their papers learns it too. The one-dimensional chain case is a special case of this, but I don’t see why, if a temporal sequence model traverses a graph (so visiting the same state for a second time would elicit the same memory pattern), it wouldn’t also pick up on its community structure (in fact, if I recall correctly, this is exactly what Schapiro’s RNN does). If the distinction is that for LAM is that the order of presentation of the memory patterns is irrelevant, then that should be highlighted, as opposed to the one-dimensional chain special case.

2. The interpretation of the sign of alpha throughout the paper seems a bit confusing without considering alpha_max or gamma – e.g. from Eq. 3, if alpha_max is very large, alpha = -1 means only local inhibition, but alpha = 0 means mostly local inhibition. Also in equation 2, the effect of alpha clearly depends on gamma. Basically, it’s unclear to me if the setpoint of zero carries any special significance (this is also evident in figure 2).

3. Relatedly, and the results section, it would be useful to clarify how the interplay between global and local inhibition produces the results, e.g. in Fig. 2. An intuitive explanation would suffice.

4. Some results in the paper, in particular figures 4, 5, and 6 are largely qualitative. It will be useful if the authors compute relevant summary statistics and perform the corresponding statistical tests.

5. Relatedly, it would be useful if if the authors put side-by-side neural data next to the simulation results in Fig 6.

6. in the bottleneck and the overrepresentation model, the additional nodes do not have the their own simulated patterns but are rather set artificially (line 591). I understand why this would be convenient for computational tractability, but it seems a bit misleading since the emergent similarity of the attractor states corresponding to those patterns is crucial for the result.

A few grammatical errors in the introduction:

L. 67 – correlations depend

L. 60 – their, mechanism

L. 83 – call our model as

L. 88 — with asymmetric

L. 90 – for hierarchical

**Have the authors made all data and (if applicable) computational code underlying the findings in their manuscript fully available?**

Reviewer #1: Yes

Reviewer #2: Yes

Reviewer #3: Yes

PLOS authors have the option to publish the peer review history of their article (what does this mean?). If published, this will include your full peer review and any attached files.

Reviewer #1: No

Reviewer #2: No

Reviewer #3: **Yes: **Momchil Tomov

Figure Files:

Data Requirements:

Reproducibility:

References:

---

## [Decision Letter · Decision Letter 1]

21 Jul 2021

Dear Dr. Haga,

We are pleased to inform you that your manuscript 'Multiscale representations of community structures in attractor neural networks' has been provisionally accepted for publication in PLOS Computational Biology.

Best regards,

Samuel J. Gershman

Deputy Editor

PLOS Computational Biology

Reviewer's Responses to Questions

**Comments to the Authors:**

Reviewer #1: I thank the authors for answering my comments and questions in detail. The manuscript has improved a lot and I believe it will be of great interest to the scientific community. I will therefore recommend its publication.

Reviewer #2: The authors have addressed all of my concerns. I recommend the acceptance of this manuscript.

Reviewer #3: The authors have addressed my comments.

**Have the authors made all data and (if applicable) computational code underlying the findings in their manuscript fully available?**

Reviewer #1: Yes

Reviewer #2: Yes

Reviewer #3: None

PLOS authors have the option to publish the peer review history of their article (what does this mean?). If published, this will include your full peer review and any attached files.

Reviewer #1: No

Reviewer #2: No

Reviewer #3: **Yes: **Momchil S Tomov

---

## [Editor Report · Acceptance letter]

17 Aug 2021

PCOMPBIOL-D-21-00434R1 

Multiscale representations of community structures in attractor neural networks

Dear Dr Haga,

I am pleased to inform you that your manuscript has been formally accepted for publication in PLOS Computational Biology. Your manuscript is now with our production department and you will be notified of the publication date in due course.

With kind regards,

Agnes Pap
